# Detection of a Stroke Volume Decrease by Machine-Learning Algorithms Based on Thoracic Bioimpedance in Experimental Hypovolaemia

**DOI:** 10.3390/s22145066

**Published:** 2022-07-06

**Authors:** Matthias Stetzuhn, Timo Tigges, Alexandru Gabriel Pielmus, Claudia Spies, Charlotte Middel, Michael Klum, Sebastian Zaunseder, Reinhold Orglmeister, Aarne Feldheiser

**Affiliations:** 1Department of Anaesthesiology and Operative Intensive Care Medicine (CCM, CVK), Charité–Universitätsmedizin Berlin, Corporate Member of Freie Universität Berlin, Humboldt-Universität zu Berlin, and Berlin Institute of Health, 13353 Berlin, Germany; matthias.stetzuhn@charite.de (M.S.); claudia.spies@charite.de (C.S.); charlotte.middel@charite.de (C.M.); 2Department of Electronics and Medical Signal Processing, Technical University, 10587 Berlin, Germany; timo.tigges@tu-berlin.de (T.T.); a.pielmus@tu-berlin.de (A.G.P.); michael.klum@tu-berlin.de (M.K.); reinhold.orglmeister@tu-berlin.de (R.O.); 3Faculty of Information Technology, Fachhochschule Dortmund—University of Applied Sciences and Arts, 44139 Dortmund, Germany; sebastian.zaunseder@fh-dortmund.de; 4Department of Anaesthesiology, Intensive Care Medicine and Pain Therapy, Evang. Kliniken Essen-Mitte, Huyssens-Stiftung/Knappschaft, 45136 Essen, Germany

**Keywords:** compensated shock, electrical cardiometry, hypovolaemia, lower body negative pressure chamber, machine learning, prediction model

## Abstract

Compensated shock and hypovolaemia are frequent conditions that remain clinically undetected and can quickly cause deterioration of perioperative and critically ill patients. Automated, accurate and non-invasive detection methods are needed to avoid such critical situations. In this experimental study, we aimed to create a prediction model for stroke volume index (SVI) decrease based on electrical cardiometry (EC) measurements. Transthoracic echo served as reference for SVI assessment (SVI-TTE). In 30 healthy male volunteers, central hypovolaemia was simulated using a lower body negative pressure (LBNP) chamber. A machine-learning algorithm based on variables of EC was designed. During LBNP, SVI-TTE declined consecutively, whereas the vital signs (arterial pressures and heart rate) remained within normal ranges. Compared to heart rate (AUC: 0.83 (95% CI: 0.73–0.87)) and systolic arterial pressure (AUC: 0.82 (95% CI: 0.74–0.85)), a model integrating EC variables (AUC: 0.91 (0.83–0.94)) showed a superior ability to predict a decrease in SVI-TTE ≥ 20% (*p* = 0.013 compared to heart rate, and *p* = 0.002 compared to systolic blood pressure). Simulated central hypovolaemia was related to a substantial decline in SVI-TTE but only minor changes in vital signs. A model of EC variables based on machine-learning algorithms showed high predictive power to detect a relevant decrease in SVI and may provide an automated, non-invasive method to indicate hypovolaemia and compensated shock.

## 1. Introduction

Haemorrhage- and fluid-loss-related complications constitute a significant cause of extended lengths of stay and deaths in hospitals [1,2]. A persistent haemorrhage or fluid loss can cause the depletion of the circulating blood volume, a condition known as hypovolemia. A hypovolaemia-associated shock develops due to a resulting inadequate circulatory flow quantified by the cardiac index (CI) and leads, depending on the length and severity of the state, to subsequent organ dysfunction [3]. Hypovolaemic shock is complicated due to the fact that the assessment of a patient’s blood volume status continues to be a difficult task for clinicians [4]. Hypovolaemic shock occurs mainly in emergency medicine, in intensive care medicine patients and perioperative situations, but on the other side can develop in any clinical situation.

To date, there are no direct measures for the circulating blood volume routinely available. Administration of intravenous fluids to treat hypovolemia is nowadays generally guided by mean arterial pressure (MAP) and heart rate (HR). However, these traditional vital signs are regarded as insensitive to hypovolaemia and can remain stable until the onset of overt conditions. The hemodynamic compensation can mask changes effectively [5] until they potentially fail abruptly at a volume loss of about 20% to 25% [6], resulting in cardiovascular collapse [7].

With a blood loss of 15% to 20% (roughly 0.75–1.25 L) of circulating blood volume a patient is—despite the nearly unchanged MAP and HR—already in a state with substantially decreased CI and stroke volume index of the heart (SVI) [8,9,10]. This stage is termed compensated shock as the circulating blood volume and circulatory flow is reduced, but MAP and HR are still maintained.

Hospitalized patients frequently develop compensated shock either due to the underlying disease, performed interventions or surgery, or by pharmaceutical interventions. Early detection of hypovolemia and compensated shock could facilitate timely treatment of volume deficits before further deterioration of the patients occur and reduce bleeding-related problems in hospitals. Clinically available tools for measuring circulatory flow are either invasive, require extended clinical training, or are too expensive for routine usage at the bedside in these patients [11,12]. Studies trying to offer an estimation of the circulatory flow parameters based on machine-learning algorithms out of non-invasive signals is up to date scarce in the literature.

Electrical cardiometry (EC) is a non-invasive method to determine CI and SVI. However, its absolute accuracy is still under debate [13,14] and only a few studies have assessed whether relative changes in SVI over time are measured accurately [15]. However, EC also measures other values to assess pre- and afterload, thoracic fluid content, and cardiac contractility. Even if the EC variables by themselves are only weakly associated with hemodynamic states, such as hypovolaemia, machine learning can combine several values to create a strong prediction of the actual hemodynamic state.

Therefore, this study aims to evaluate the ability of different machine-learning algorithms using EC-derived parameters to predict a decline in SVI greater than or equal to 20% compared to the baseline measured by transthoracic echocardiography.

The cut-off was selected to detect a state of compensated shock, characterized by a decline in SVI, but also MAP and HR as the usually determined parameters, to evaluate if the circulatory status is still unchanged. This state is frequently missed by clinicians, but already exposes the patients to reduced organ perfusion and subsequent risk of organ dysfunction.

Further goals are to compare these models’ performances to the predictive power of the vital signs HR, systolic (SBP) and diastolic arterial blood pressure (DBP), and MAP, as well as to the EC-derived measurements, each taken separately.

## 2. Materials and Methods

This study was a prospective single-centre, experimental trial in healthy volunteers conducted as a pilot study. Ethical approval for this study (Ethical Committee No. EA1/249/17) was provided by the ethics committee of the Charité—Universitätsmedizin Berlin (Chairperson Prof. R. Dr. med. Morgenstern). The trial was registered internationally (ClinicalTrials.gov Identifier: NCT03481855, principal investigator: Aarne Feldheiser, date of registration: 5 March 2018).

All volunteers were checked for eligibility and gave their informed, written consent. The study protocol was carried out in accordance with the Declaration of Helsinki. The trial was performed from 6 March 2018 to 11 April 2018 at the Charité-Universitätsmedizin Berlin, Campus Virchow-Klinikum, Berlin. The project included several research questions to minimize the risks of volunteers and reduce research funds. The first publication of this project covered methodological aspects [16].

Participants: Test subjects had to be male, at least 18 years of age, and had to have a valid German health insurance. Exclusion criteria were age > 40 years, acute illness, known chronic cardio-vascular (especially arterial hypertension, pacemakers, defibrillators), renal, pulmonary, neurological, metabolic, or gastrointestinal diseases, as well as clinical signs of reduced cardiorespiratory capacity. Anamnestic hints for syncope or predisposition to hypotension were also reasons for exclusion. Further exclusion criteria were the presence of inguinal hernias and the intake of any long-term medication and the refusal of the storage of pseudonymized data collected during the study. Any volunteer showing relevant pathologies in the baseline physical examination or TTE was excluded. The volunteers were blinded to the results displayed on measuring devices and data collection monitors and received only the results of the initial TTE. There was no financial compensation for participation.

Study Protocol: After the patients gave their informed written consent, baseline demographic data were recorded, and the volunteers were introduced to the laboratory equipment. Then, the volunteers were placed within the LBNP chamber in the supine position with a slight tilt of 20° to 30° to the left using positioning pillows to facilitate the TTE exam. Their lower body laid within the chamber and a neoprene kayak skirt around their waist sealed the opening of the chamber. An adjustable bike saddle inside the chamber prevented the subject from being pulled further inside when negative pressure was applied and reduced the physical strain on the leg muscles to maintain positioning within the chamber. The LBNP chamber was constructed according to recent recommendations [17,18] using solid wood for the chamber frame and plywood sheets for the top, bottom, and side walls. It was sealed with commercially available silicone gel. The simulated progressive central hypovolaemia of the LBNP chamber was haemodynamically and humorally extensively characterized, and the protocol was previously published by our group [16]. Negative pressure was applied by fitting the hose of a commercially available vacuum cleaner (VS06B1110, Siemens AG, Berlin and Munich, Germany) into a hole in the chamber wall. The amount of negative pressure was regulated by adjusting the power of the vacuum cleaner. Pressure could be regulated continuously between 0 mmHg and −60 mmHg, and precise regulation could be achieved by the opening and closing of valves integrated into the front panel. The negative pressure within the chamber was monitored with a digital pressure measuring instrument (PDA, PCE-Deutschland, Meschede, Germany).

After the acquisition of the initial TTE and the establishment of all monitoring devices, a ten-minute baseline measurement at 0 mmHg LBNP was performed. Subsequently, progressive central hypovolaemia was induced at stage 1 with −15 mmHg, at stage 2 with −30 mmHg, and at stage 3 with −45 mmHg chamber pressure for seven minutes each. Recovery (return to 0 mmHg chamber pressure) was monitored for ten minutes. According to Cooke and colleagues [10], a pressure of −15 mmHg simulates a mild blood loss of approximately 10 %, and −30 mmHg simulates a moderate blood loss of approximately 10–20% total blood volume. The final stage at 45 mmHg negative chamber pressure simulates a severe blood loss of approximately more than 20% loss of total blood volume [10]. A summary of the study protocol is offered in Figure 1.

Data collection: Prior to the study protocol, a complete TTE exam was performed. All cardiac valves, chambers, and the right and left ventricular systolic function were evaluated to exclude underlying or undetected pathologies, which would lead to the exclusion of the volunteer. The measurements were performed on a Vivid S70 (GE Healthcare, Boston, MA, USA) by an examiner (A.F.) certified by the European Society of Cardiology/European Association of Cardiovascular Imaging for adult transthoracic echocardiography.

During the study protocol, the volunteers’ HR was monitored by an electrocardiogram with standard Eindhoven leads. Systolic (SBP), diastolic (DBP), and mean arterial pressure (MAP) were measured by an automated, oscillometric measurement device (BSM-3000, Nihon Kohden Europe GmbH, Rosbach, Germany), with the pressure cuff placed around the volunteers’ upper right arm once one minute before the end of every LBNP level.

SV was determined within the last two minutes of every time interval. Briefly, the diameter of the annulus of the left ventricular outflow tract (LVOT) was measured during early systole in the parasternal long-axis (PLAX) view, and the cross-sectional area (CSA) was calculated. The left ventricular outflow velocity was recorded by pulsed-doppler from the apical five-chamber view at the annulus of the LVOT in line with the outflow direction, and the velocity-time-integral (VTI) determined from the systolic flow signal. SV was then calculated by multiplying CSA by VTI. For the various calculations of SVI at the different stages, the LVOT diameter of the baseline measurement was used. Measurements at the different stages revealed that the LVOT diameter did not change significantly from the baseline values. During the last two minutes of every LBNP pressure level, a median of nine VTI measurements was taken.

The study was part of a full-scale echocardiographic project for the assessment of hypovolaemia. For this approach, all volunteers received a standardized, comprehensive transthoracic echo exam at every stage of the study protocol. Intra-observer agreement and variability as well as inter-observer agreement and variability and Bland–Altman plots demonstrated in a similar way good intra- and inter-observer agreement; but, as a limitation, it must be stated that the measurements of agreement were not performed for SVI.

Electrical cardiometry parameters were recorded continuously by an Osypka ICON™ monitoring device (Osypka Medical, Berlin, Germany). Four electrodes were placed according to the manufacturer’s instructions: the first pair of electrodes on the left lateral base of the neck and 10 cm above, and the second pair on the left medial axillary line, approximately at the same height as the xyphoid process and 10–15 cm below. A table of the determined EC parameters displayed by the ICON™ monitoring device is listed in Table 1. The device recorded data at a rate of 200 Hz and saved it in a beat-to-beat fashion. The data collected within the last two minutes of each step of the experiment was used for further transformation and analysis.

Statistical analysis and determination of model performance: All data validation, transformation, and analysis were performed using the programming language R for statistical computing (Version 3.5.1) [20] and the software R Studio^®^ (RStudio PBC, Boston, MA, USA, Version 1.2.1335). Model generation was performed using the caret package for R (Version 6.0-86) [21].

Because of limited sample sizes and observations not being normally distributed, data were expressed as the median (25%; 75% quartiles) or frequencies (%), respectively. A paired Wilcoxon signed-rank test was conducted for each recorded parameter and LBNP phase to assess whether its distribution differed significantly from the baseline recordings. Additionally, the relative changes in each parameter with regard to its respective baseline value were calculated to examine the subject-specific trends in the data. Group differences between the AUC of the vital parameters and machine learning were evaluated using unpaired non-parametric testing.

To assess the correlation between LBNP-induced SV changes and alterations in the other measured bioimpedance features, a correlation analysis was conducted. First, the population-wide correlation was evaluated by calculating both the Pearson (ρP) and Spearman (ρS) correlation coefficient for all collected data points. Next, these correlation coefficients were also calculated for each subject individually, aiming to determine the intra-individual correlations. When assessing the correlations, the definition of correlation strength by Cohen was used.

To assess the diagnostic value of the vital signs and selected bioimpedance parameters with respect to the binary outcome variable of an SV-TTE decrease greater 20%, ROC curves, including a grey zone approach, were calculated [22]. This approach avoids dichotomizing the population, as it provides two cut-offs: the lower cut-off of the grey zone rules out an SV-TTE decrease of more than 20% with near certainty, and the upper cut-off rules it in [22,23]. Consequently, this method allows clinical decision making with respect to an SV-TTE decrease of more than 20%. First, the AUC and 95% CI were computed by averaging 1000 bootstrap samples from the study population. Second, the best cut-off was determined maximizing the Youden index [24]. The best cut-off determination was conducted for 1000 populations bootstrapped from the study population and the mean value and its 95% CI were then estimated. Finally, the grey zone was defined as the 95% CI of the best cut-off [23]. For visualization, the grey zone was included into boxplots comparing the vital signs and selected bioimpedance parameters stratified by an SV-TTE decrease of more than 20%. A two-tailed *p*-value < 0.05 was considered statistically significant.

The mean, median, standard deviation (SD), variance (var), skewness (skew), kurtosis (kurt), and interquartile range (IQR) values were calculated of the last two minutes before every LBNP level change for every parameter of the Osypka ICON™ monitoring device. The data were transformed into z-scores for all methods of model generation using the carets “center/scale” function. These transformed values were taken as feature inputs to the models.

To maximize the number of observations from a limited amount of test subjects, we related each observed value (transformed in the way explained in the previous paragraph) to all its previous observations in the same subject. For example, an observation from pressure stage 4 was related to the observations at baseline and stage 1, 2, and 3. An observation at stage 2 was only related to the observations of the same value at baseline and stage 1. The baseline measurement was included once without relation to the pressure stages. This resulted in a theoretical maximum of eleven relational observations in 30 test subjects equal to 330 relational observations for analysis.

A decrease in SV-TTE of 20% or higher was considered as the threshold for a relevant reduction in circulatory flow [3] and therefore set as a binary outcome variable for machine-learning modelling and model prediction. We chose several widely used classification methods (k-nearest neighbours (KNN), Naive Bayes (NB), Random Forest (RF), and Support Vector Machines using a radial (SVMrad) or a linear kernel (SVMlin)), which usually perform well with few observations of many features.

A visualization of the process of model generation described below is given in Figure 2. It was decided to use a test-subject-based leave-two-out cross-validation (LTOCV) to assess the generalization capabilities of the machine-learning approach. Specifically, data from two randomly selected test subjects were held back as a testing dataset, while data from the 27 remaining test subjects were used as a training dataset in each LTOCV fold. This process was repeated 29 times to average out the influence of the selection of certain test subjects’ data as a training or test dataset. This specific data split was chosen as it showed an acceptable bias-variance trade-off for the estimation of model performance in this limited dataset.

Within each LTOCV fold, feature selection was conducted by a forward wrapper approach, where the models’ hyperparameters were tuned by performing a grid search. An overview of the feature selection and hyperparameter grid search are displayed in Table 2. Ten-fold cross-validation estimated performance values are considered in this model optimization process. This combination of two cross-validation loops is commonly referred to as nested cross-validation [25]. Grid search hyperparameters and their corresponding performance were visualised during algorithm development. If the best performance values were found at the edges of the grid search, the grid search was manually expanded, and model generation repeated.

The defined primary outcome value was the area under the curve of a receiver operating characteristics curve (AUC), which analyses the predictive power of the model in discriminating between a decrease in SV-TTE of ≥20% vs. <20%. The AUC for each of the 29 iterations of the cross validation was calculated and then used in the final evaluation of the algorithm’s performance. Secondary outcome variables for assessing the accuracy and predictive power of EC were the AUC for singular EC measurements values and vital signs in predicting SV decrease, as well as the correlation coefficients for the EC values to SV measurements by TTE.

## 3. Results

Characteristics of the healthy volunteers: 30 subjects volunteered to be part of this study. Data from 29 test subjects were analysed. In one test subject, data acquisition failed due to technical problems. In 34 of 319 relational observations (11%), SV-TTE was not acquired in one or both compared timepoints; they had to be excluded from the analysis. For absolute observations of measurement values, data were missing in 3 of 145 (2%) cases.

The median age of the subjects was 27 years (IQR 25 to 75), the median body height was 183 cm (181 cm to 186 cm), and the median body weight was 79 kg (72.5 kg to 89.3 kg). There were no comorbidities reported, and in the initial TTE examinations, no pathologies were detected.

Hemodynamic changes during LBNP: The relative changes to the baseline measurement of SV-TTE, vital signs, and selected EC measurement values are shown in Figure 3. SV-TTE compared to the baseline decreased further with each increment of negative pressure (during the last pressure increment, at −45 mmHg: median decrease to 65.7% compared to baseline measurement (IQR 56.9% to 72. 2%, *p* < 0.001)). SV, CO, and the ICON measured by EC significantly decreased at all pressure levels and remained reduced during recovery. The decrease was the strongest when the highest amount of negative pressure was applied. At −45 mmHg, SV-EC showed a median decrease to 79.9% of the measured baseline values (IQR 73.8% to 84.2%, *p* < 0.001). CO was reduced to 89.5% compared to the baseline (IQR 81.9% to 95%, *p* = 0.037). Relative ICON values were decreased to 70.4% (IQR 59.6% to 78.2%, *p* < 0.001). Notably, the mean HR decreased slightly during −15 mmHg LBNP (median decrease to 96.9% compared to the baseline value (IQR 94% to 100%,, *p* = 0.002) and recovery (decrease to 92% compared to baseline (IQR 89.3% to 94.8%, *p* < 0.001) and only increased at lower negative pressure levels (−30 mmHg (median increase 102.2% (IQR 99.9% to 105.5%, *p* = 0.034)) and −45 mmHg (median increase 109.5% (IQR 105.6% to 114.9%, *p* < 0.001)). While the MAP stayed stable throughout the experiment and only increased slightly during recovery (102% compared to baseline (IQR 99.1% to 106.4%, *p* = 0.006)), SBP significantly decreased throughout the application of −30 mmHg (95.3% of baseline measurement, IQR 93.5% to 96.8%, *p* < 0.001) and −45 mmHg (94.8% of baseline measurement, IQR 93.1% to 98.7%, *p* < 0.001). DBP increased slightly during −45 mmHg (102.7%, IQR 100.3% to 104.7%, *p* = 0.003) and recovery (103.8%, IQR 99.7% to 106.1%, *p* = 0.002). In this trial population, 1 of 142 absolute observations (0.7%) showed an absolute HR ≥ 100 min^−1^. No observations with an SBP ≤ 100 mmHg or a MAP of ≤60 mmHg were recorded. In contrast to that, 85 out of 285 (28.1%) of the relational observations showed significant hypovolaemia (decrease of SV > 20% compared to baseline) measured by TTE.

Correlation of changes in hemodynamic features to a decrease in SV: Exploring the correlation of the features to the changes of SV-TTE, a strong positive correlation can be observed for SV measured using EC (SV-EC, ρP = 0.856 (IQR 0.652 to 0.936)). Compared to SV-EC, the vital signs showed weaker correlations, the strongest being SBP (ρP = 0.624 (IQR 0.324 to 0.817)) and HR (ρP = −0.576 (IQR −0.795 to −0.284)). Regarding the EC-derived features, strong correlations were found for ICON (ρP = 0.806 (IQR 0.542 to 0.921)), heart rate variability (HRV) (ρP = 0.797 (IQR 0.096 to 0.871)), pre-ejection period (PEP) (ρP = −0.839 (−0.948 to −0.757)), systolic time ratio (STR) (ρP = −0.902 (IQR−0.941 to −0.72)), and stroke volume variation (SVV) (ρP = −0.706 (IQR −0.887 to −0.366)). Intra-individual correlation coefficients of the most prominent features are shown in Figure 4.

Predictive power of single features for the detection of a decrease in SV: Features showing a strong correlation to changes in SV-TTE were further explored (HR, SBP, SV, PEP, STR, and ICON). We calculated a ROC curve for each feature predicting a decrease in SV-TTE ≥ 20% compared to the baseline. ROC and its AUC of the vital signs and selected EC-derived features can be seen in Figure 5.

Grey zones discriminating a stroke volume decrease above and below 20% compared to the baseline (SV-TTE) for the same selected features as for the ROC analysis are further displayed in graphs provided in Figure 6). The highest AUC was reached by PEP (AUC = 0.79 (95% CI 0.71 to 0.86)) and STR (AUC = 0.79 (95% CI 0.7 to 0.86)). Changes in SV-EC achieved an AUC of 0.72 (95% CI 0.63 to 0.81). The change in HR showed an AUC of 0.72 (95% CI 0.62 to 0.81) but had a strikingly small grey zone (higher and lower level: 1.013 and 1.003).

Feature generation and characteristics and predictive power of machine learning algorithms: In total, 109 features were calculated from the EC values based on descriptive statistical measures. The features underwent feature selection for every model in the nested cross-validation scheme. Figure 7 depicts which features were used by the different supervised learning methods. The median number of features used was 40. KNN always used all 109 offered features.

All machine-learning methods significantly outperformed DBP (AUC: 0.58 (95% CI 0.51 to 0.70)) and MAP (AUC: 0.58 (95% CI 0.55 to 0.6)) in predicting a decrease in SV-TTE. Models generated by SVM with a linear kernel had a statistically significant higher AUC (0.86 (95% CI 0.78 to 0.96)) than SBP (AUC: 0.82 (95% CI 73.7 to 0.85)), but not HR (AUC: 0.83 (95% CI 0.73 to 0.88)). Random Forest was the most accurate method explored for predicting a decrease in SV-TTE, showing a significantly higher AUC than all vital sign features (AUC: 0.91 (95% CI 0.83 to 0.94)).

## 4. Discussion

During the simulation of shock by the LBNP chamber, the volunteers presented a substantial decrease in SV-TTE at all stages, reaching a median decrease of 34.3% at the highest stage while the commonly monitored vital signs—namely HR and MAP—revealed only minor changes (12.5% and 5.2%, respectively) and were within the normal physiological limits. Considering that a negative pressure of −45 mmHg simulates a blood loss of 20% or higher [10], it must be stated that in contrast to the substantial decrease in SV-TTE, in our study, the vital signs were not able to reveal the extent of central hypovolaemia. The main finding of this study is that an SV-TTE decline of 20% or more, as a marker of compensated shock, can be detected with high accuracy by a machine-learning algorithm (Random Forest) based on EC features, and is superior in accuracy to changes in vital signs.

The lower body negative pressure (LBNP) chamber is a well-established method in research and medical education to simulate bleeding by reversibly pooling blood to the lower extremities [26]. According to our data presented here, even at low negative pressure levels, a decrease of SVI and CI can be shown. In experiments, haemodynamic [19,20,26] and neuro-humoral [27] responses were like induced blood loss. It was even possible to directly associate negative pressure levels with the amount of blood loss [19]. Johnson et al. further showed that from a negative pressure of −13 mmHg, all volunteers in their experiment showed a reduction in splanchnic perfusion and a corresponding rise in calculated splanchnic vascular resistance [28]. A reduced splanchnic perfusion of the vital abdominal organs, such as the liver, stomach, and small and large intestine, instantly is associated with a reduced organ function, exposing the patient at risk for decompensation or complications. Due to our data presented here and in combination with the literature, we consider the LBNP chamber appropriate to simulate compensated shock in this study.

Selected EC variables (e.g., SV-EC, STR, and PEP) promisingly showed a relative change in their values compared to the baseline, matching the relative decline in SV-TTE. This was supported by high correlation coefficients and findings from validation studies [29,30]. However, the moderate to low AUC values are contradictory. Therefore, it is important that the broad ranges of the upper and lower limits of the grey zones, when discriminating an SV-TTE decline of ≥20%, confirm the missing precision, as they indicate a low to moderate prediction ability of the single EC variables. Comparing the time course of for example SVI-TTE to SVI-EC offers a possible explanation, as during the baseline conditions and extreme changes in SVI-TTE the parameter SVI-EC shows a high correlation, but the decline during the initial stages is only weekly correlated. According to a recent meta-analysis [31], EC is not able to replace thermodilution and TTE for the measurement of absolute CO values nor the trending ability. In our study, the employed machine-learning methods, especially the Random Forest algorithm, were able to integrate the individual “weak learners” into a framework that achieved a high prediction ability.

Our observations confirm previous findings regarding the characteristics of simulated and experimental bleeding. For example, progressive bleeding in dogs indicated a substantial decrease in SV, whereas MAP was maintained until approximately 20% of blood loss was reached [3]. The previously published data of the simulated central hypovolaemia by the LBNP chamber is well in line with these results of experimental bleeding [9]. Hamilton-Davies and colleagues showed with bleeding experiments in healthy volunteers (blood loss ≥ 1.5 L) that HR and MAP did not indicate the extent of the blood loss, while SV decreased. The response of HR in the grey zone analysis from our data misleadingly indicates a good discriminating ability for the detection of the decline in SV-TTE ≥ 20%. However, an increase of 1.3% must be considered clinically irrelevant.

Based on these previous findings [9], we chose an SV-TTE decline of ≥20% compared to the baseline as the target value for training the machine-learning algorithms, to detect compensated shock prior to the development of pathological vital signs or symptoms indicating the clinically obvious decompensated shock, for which RF showed the best predictive power.

Convertino and colleagues published results regarding the prediction of decompensation of healthy volunteers due to hypovolaemia using machine learning. They described the process of image-based feature extraction from observed waveforms recorded by different non-invasive monitoring methods in 104 subjects during LBNP experiments [32]. The generated model predicted the level of LBNP (and therefore, the level of simulated central hypovolaemia) very accurately. In this regard, their work is very similar to ours in goals, methods, and results. However, the process of the actual feature extraction and machine-learning modelling has not been explained thoroughly.

Outlook: Non-invasive screening for compensated shock machine-learning methodologies in the prehospital and perioperative setting as well as in the intensive care unit might lead to improved allocation of resources, as physicians could direct advanced hemodynamic monitoring and therapy to patients needing it based on objective measurements.

Limitations: It is to note that the calculated AUC for HR and blood pressure in Figure 7 differ from those initially calculated in Figure 5. The reason for this difference is that the first calculation was based on the whole dataset while the second used only the much smaller test dataset partitions in the cross-validation scheme. The difference in the AUC estimates might indicate that the variance in the randomly selected test data does not adequately represent the entire data set. More data would lead to more representative and less variant test data partitions.

Due to the relatively small sample size, a compromise between the fraction of test and training data had to be found during the design of the algorithm, as explained above. However, vital signs and machine-learning models should be equally affected by this effect so that we can assume a similar relationship between the AUC when theoretically applied to the whole dataset.

This work is based on a small experimental pilot study in healthy men. Further studies are needed to improve its clinical value. Consequently, the model should be applied to more clinically relevant patient populations of men and women alike, perioperatively, in emergency medicine, or intensive care medicine.

## 5. Conclusions

A framework based on models generated with machine-learning algorithms using EC-derived features was able to detect an SV decline of 20% or more with high accuracy during central hypovolaemia. RF and SVMlin showed the best performance in predicting this decline amongst the chosen modelling methods. The results for the RF algorithm are superior to any single EC feature or vital sign. A high model performance was reached although taken separately. The EC features entering the algorithm did not show comparable accuracy. This study describes the possibility to develop a new approach to detect compensated shock by using supervised machine-learning methods.

## Figures and Tables

**Figure 1 sensors-22-05066-f001:**
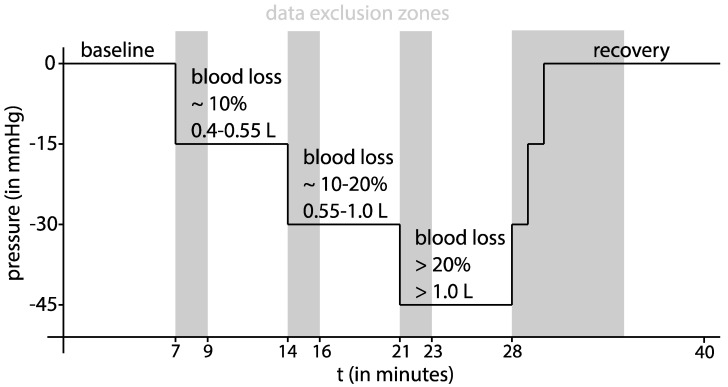
Time course of the LBNP study protocol: the expected simulated blood loss is given in accordance to [19]. The first two minutes of each LBNP stage and the first seven minutes of the recovery stage were excluded from any further analysis. These data exclusion intervals are marked in grey.

**Figure 2 sensors-22-05066-f002:**
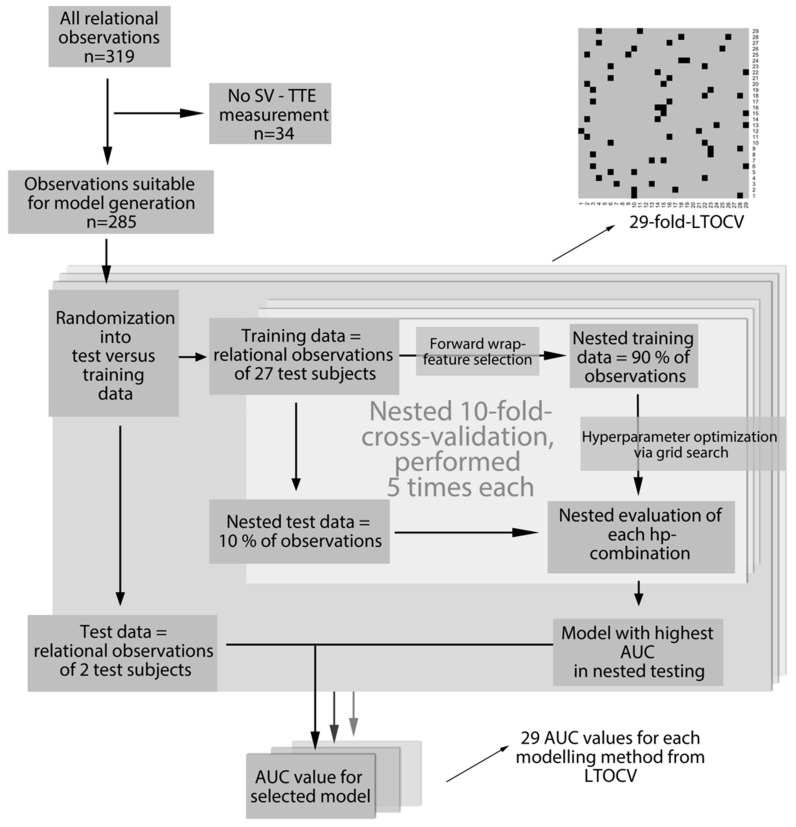
Machine-learning algorithm: schematic overview of the model generation process. At the top right, the grid shows the subject numbers on the x-axis, and the repetition of the “outer” test-subject-based LTOCV on the y-axis. Every repetition of cross-validation had two subjects’ data (marked with a black square) held back for testing of the model’s performance. The remaining data were used for model generation. AUC = area under the curve; hp = hyperparameter; LTOCV: leave-two-out cross-validation; SV-TTE: stroke volume measured by transthoracic echocardiography.

**Figure 3 sensors-22-05066-f003:**
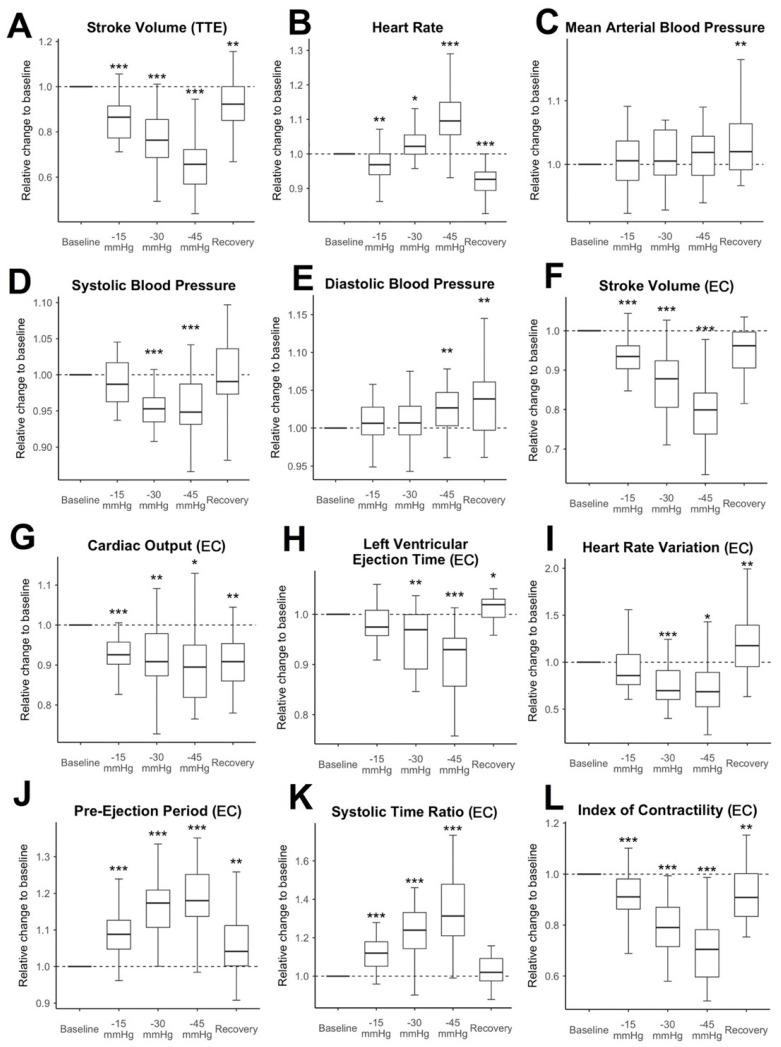
Time course of changes in the features relative to baseline: changes to the selected hemodynamic measurement values (one panel per value) over the course of the LBNP experiment relative to the subjects’ individual baseline measurement. The subfigures (**A**–**L**) describe Stroke Volume determined by TTE (**A**), the vital signs (**B**–**E**), and selected parameter determined by EC (**F**–**L**). The median baseline value is marked with a dashed line. Results of paired nonparametric tests compared to the baseline are indicated as follows: * = *p* ≤ 0.05; ** = *p* ≤ 0.01; *** = *p* ≤ 0.001. (EC): indicates values determined by electrical cardiometry; (TTE): indicates values determined by transthoracic echocardiography.

**Figure 4 sensors-22-05066-f004:**
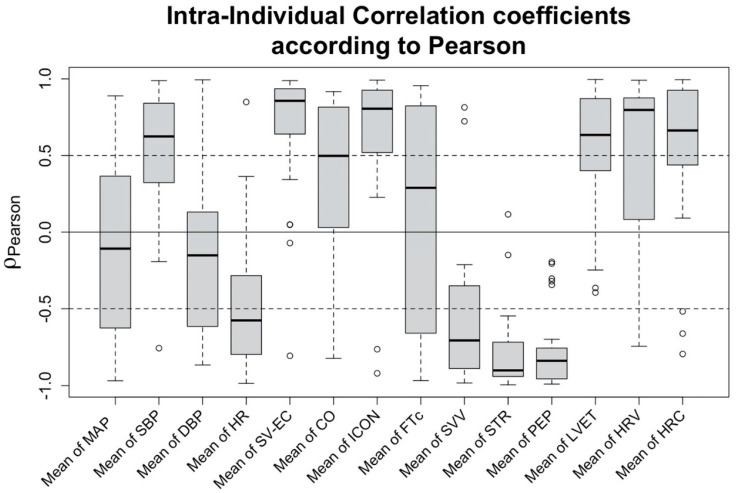
Boxplots of the correlation between the selected features and TTE-SV: Intra-individual correlation (according to Pearson) between haemodynamic values and SV-TTE. Each boxplot represents 29 correlation coefficients, one for each test subject. Black rings represent outliers.

**Figure 5 sensors-22-05066-f005:**
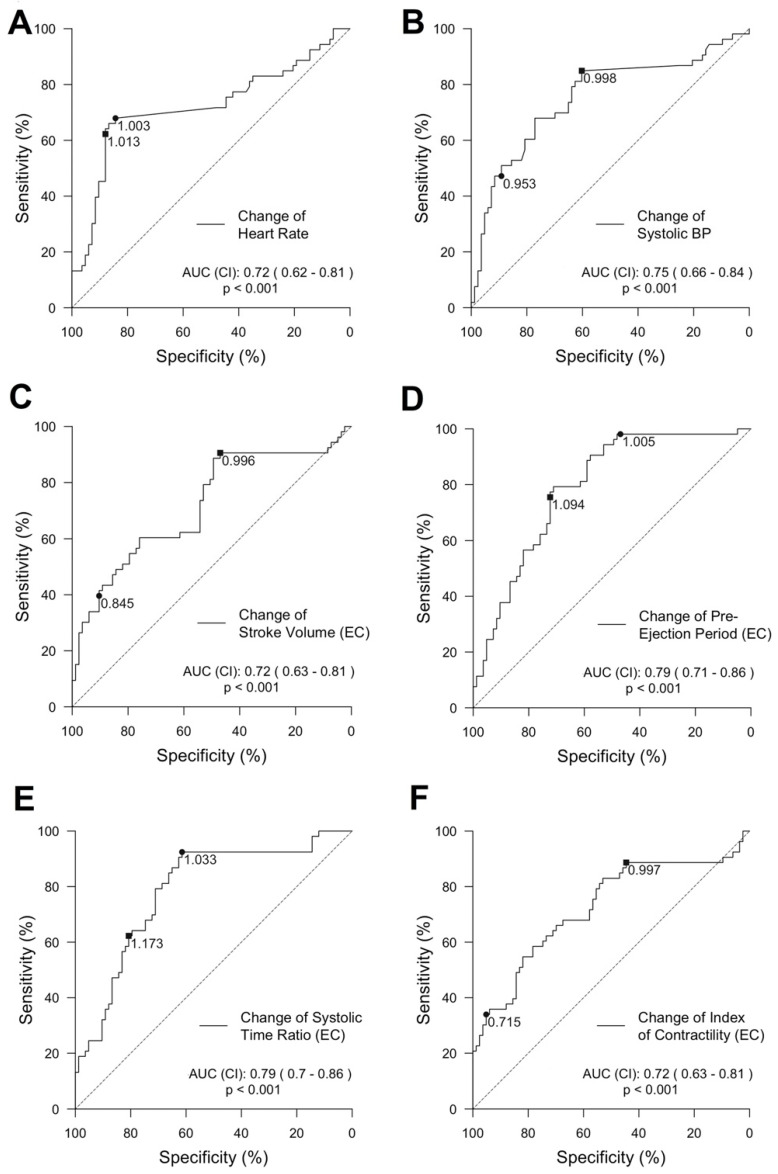
ROC curves and grey zones of the selected features: ROC curves in predicting a decrease in SV-TTE for the selected hemodynamic values (one per panel). The subfigures (**A**–**F**) describe the change of the vital signs Heart Rate and Systolic BP (**A**,**B**) and selected parameter determined by EC (**C**–**F**). ROC curves are displayed in the usual fashion. The area under the curve (ROC-AUC) and its confidence intervals (CI) due to bootstrapping can be found at the bottom right of each plot. The cut-off values of the diagnostic grey zone are shown on the ROC curve. (EC): indicates values determined by electrical cardiometry; BP: blood pressure.

**Figure 6 sensors-22-05066-f006:**
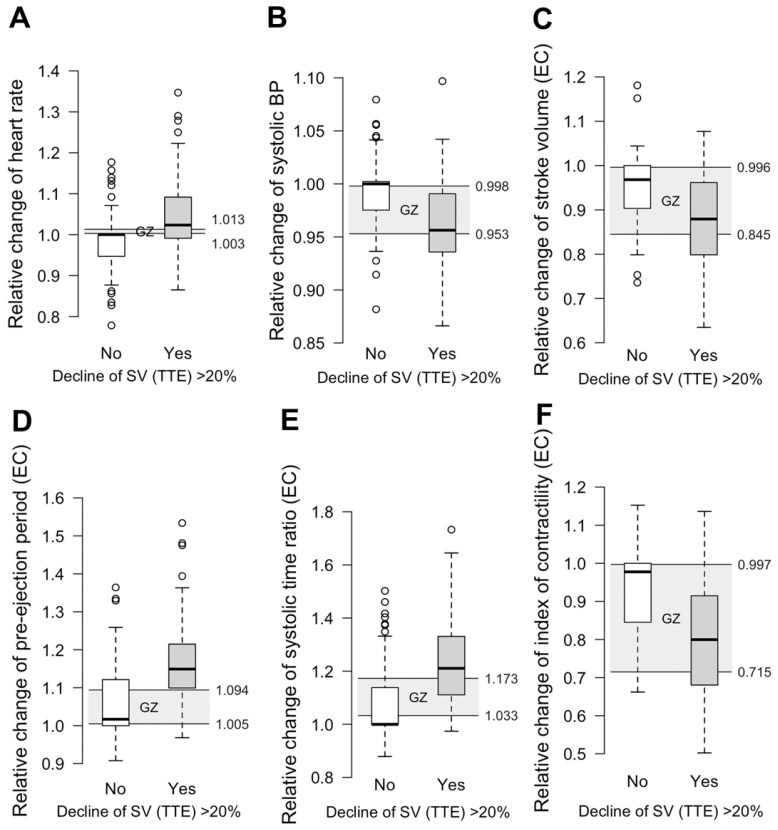
Discrimination grey zones of the selected features: visualization of the grey zone in predicting a decrease in SV-TTE for the selected hemodynamic parameters. The subfigures (**A**–**F**) describe the change of the vital signs Heart Rate and Systolic BP (**A**,**B**) and selected parameter determined by EC (**C**–**F**). Boxplots show the distribution of observations between the two groups (decline in SV-TTE ≥ 20%/no decline). A diagnostic grey zone was calculated to show the area in which the observed values cannot distinguish between the two groups. EC indicates the parameters determined by EC using thoracic electrical bioimpedance. Black rings represent outliers.

**Figure 7 sensors-22-05066-f007:**
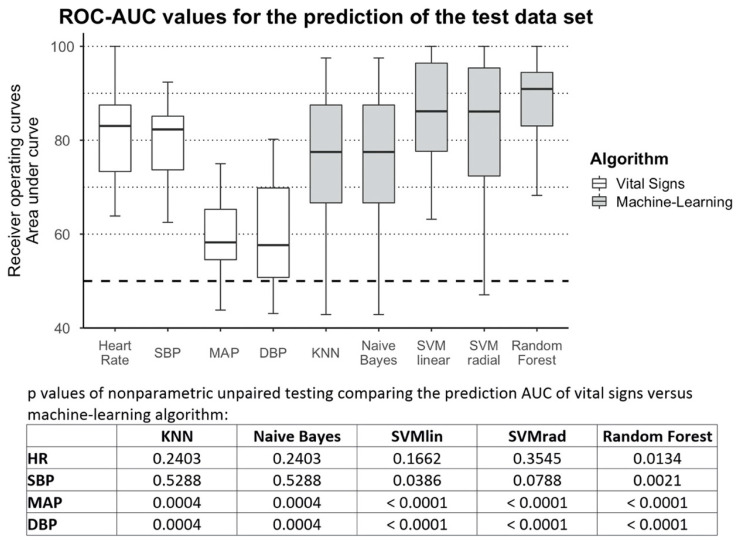
Machine-learning prediction results and statistical comparison: boxplots of the area under the curve (ROC-AUC) of the vital signs and generated models when predicting the stroke volume decrease measured by transthoracic echocardiography (SV-TTE) in the test dataset. Results of unpaired non-parametric testing for comparing results are displayed in the table below. DBP: diastolic blood pressure; HR: heart rate; KNN: K-nearest neighbours; MAP: mean arterial pressure; SBP: systolic blood pressure SVMlin: Support Vector Machines using a linear kernel; SVMrad Support Vector Machines using a radial kernel.

**Table 1 sensors-22-05066-t001:** Overview of the electrical cardiometry parameters recorded continuously by the Osypka ICON™ monitoring device.

Abbreviation	Name of Parameter	Unit	Definition
SV	Stroke Volume	mL	Blood volume ejected from the left ventricle during systole
HR	Heart Rate	min^−1^	Cardiac cycles/Minute
CO	Cardiac Output	L × min^−1^	SV/HR
SI	Stroke Index	mL × m^−2^	SV/body surface area
CI	Cardiac Index	L × min^−1^ × m^−2^	CO/body surface area
PEP	Pre-Ejection Period	Ms	Time period from beginning of the chamber complex measured by ECG to ejection of blood from the left ventricle
LVET	Left Ventricular Ejection Time	Ms	Duration of systolic blood ejection
FTC	Corrected Flow Time	Ms	Frequency corrected LVET using Bazett’s formula
STR	Systolic Time Ratio		PEP/LVET
ICON	Index of Contractility		Peak acceleration of erythrocytes in the aorta, calculation described in 16
VIC	Variation in Contractility	%	Variability of ICON
SVV	Stroke Volume Variation	%	Variability of SV
HRV	Heart Rate Variability	Ms	Variability of R-R Intervals in EKG analysis
HRC	Heart Rate Complexity		Heartbeat complexity analysis using sample entropy analysis
MSE	Multiscale Entropy Complexity Calculation		Analysis of self-similarity between signals
PNN	Interval-based Distance Ratio Calculation	%	Calculation of pNN50 as a surrogate parameter for parasympathetic activity 19
STAT	Signal Stationarity Calculation		Index calculation StatAv proposed by Pincus and colleagues
CCC	Cardiac Cycle Counter	mL	Cardiac cycle counter for development purposes

**Table 2 sensors-22-05066-t002:** Overview of the feature selection and hyperparameter grid search of the machine-learning algorithms.

Machine-Learning Algorithm	Initial Grid Description	Final Grid Description	Highest Performing Feature Selection Method (R-Package)
**k-nearest neighbours (KNN)**	k = (1–105 increasing in increments of 2)	k= (1–109 increasing in increments of 2)	All features used
**Naïve Bayes (NB)**	adjust = (0.5, 1, 1.5) Kernel use = (FALSE, TRUE) fL = (0, 1)	adjust = (−1.5, −1, −0.5, 0, 0.5, 1, 1.5) Kernel use = (FALSE, TRUE) fL = (−1, −0.5, 0, 0.5, 1)	Recursive feature elimination (caret)
**Support Vector Machine using radial kernel (SVM rad)**	Sigma = (2^−9^, 2^−8^, 2^−7^, 2^−6^, 2^−5^, 2^−3^) Cost = (0.4, 0.6, 0.8, 1, 2, 2^2^, 2^3^)	Sigma = (2^−10^, 2^−9^, 2^−8^, 2^−7^, 2^−6^, 2^−4^) Cost = (0.1, 0.2, 0.4, 0.6, 0.8, 1, 2, 2^2^, 2^3^, 2^4^)	gain.ratio (FSelector)
**Support Vector Machine using linear kernel (SVM lin)**	Cost = (0.001, 0.01, 0.2, 0.4, 0.6, 0.8, 1, 2, 4)	Cost = (10^−4^, 10^−3^, 10^−2^, 0.1, 0.2, 0.4, 0.8, 1, 2, 4)	Combination of features selected by all 3 entropy-based FSelector methods
**Random Forest (RF)**	mtry = (1–15 increasing in increments of 1)	mtry = (1–15 increasing in increments of 1)	Combination of features selected by all 3 entropy-based FSelector methods

## Data Availability

Due to data restrictions the original data cannot be provided.

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
