# Peer review of "Detection of a Stroke Volume Decrease by Machine-Learning Algorithms Based on Thoracic Bioimpedance in Experimental Hypovolaemia"

_sensors, 2022, doi:10.3390/s22145066_

Round 1

Reviewer 1 Report

Studies on the new non-invasive ways of hemodynamic status monitoring and its changes in ICU setting are of potentially high clinical significance. Therefore I have read with interest the study by Stetzuhn et al. where the authors analyze the use of data derived from thoracic bioimpedance for machine-learning algorithms to detect stroke volume decrease in a simulated hypovolemia model. Hypovolemia was simulated by lower body negative pressure chamber in 30 healthy male volunteers. Stroke volume decrease was assessed by means of transthoracic echocardiography. The study is well presented, but I have a few methodological comments, which in my opinion should be addressed as they limit the value of the results.

1. First of all the stroke volume decrease, as stated above, was assessed indirectly by TTE. However, it was done by a single experienced operator only (lack of inter-personal variability). I also do not see the intra-personal variability analysis. Please comment or add to limitations. 

2. Secondly, thoracic bioimpedance is know to have limited repeatability. It seems that the results were based on single readouts and not on repeated measures. Is there any reported repeatability analysis in the literature for the specific machine used? Please address or add to limitations if there is no data. 

Author Response

1. 

We thank the reviewer for this remark. The study was part of a full-scale echocardiographic project for the assessment of hypovolaemia. For this approach all volunteers received a standardized, comprehensive transthoracic echo-exam at every stage of the study protocol. Within that echocardiographic project the measurements were repeated by the first observer and additionally checked by a second observer without knowledge of the results of the first observer. The assessment of intra-observer agreement and variability showed an intra-class correlation coefficient (ICC) of 0.977 (confidence interval [CI] 0.968–0.983) and the absolute difference (in %) divided by the mean of the repeated measures of 2.9%, but this was for the parameter TAPSE and not for SV or VTI. Regarding inter-observer agreement, the ICC, and the absolute difference (in %) divided by the mean of the repeated measures were 0.976 (CI 0.957–0.987) and 3.9% for TAPSE and Bland–Altman plots demonstrated in similar way good intra- and inter-observer agreement.

Although these data were not for SV or VTI, the data show that we reached a high intra- und inter-observer agreement, but as we don’t show TAPSE values in the manuscript we decided to leave it out.

Considering your point, we added a sentence to the results section that we generally within that study we had high intra- und inter-observer agreement but as a limitation we didn’t do it on SV.

2. 

Regarding the measurement of the bioimpedance values we used a two-minute window during every stage to increase validity of data. During the study conductance we observed only slight undulations of bioimpedance values, but no relevant instability of measurements, and the analysis of data didn’t reveal a continuous change (increase or decline) over the two-minute windows. Consequently, we do not believe that we had an issue of limited repeatability.

Reviewer 2 Report

Dear authors,

I found the article very well written and interesting to read. I have only 2 very small remarks.

Line 49: delete too..

Table 2: add explanation for abbreviations.

Author Response

We thank you for your comments and we changed the manuscript accordingly.

Reviewer 3 Report

The authors propose a prediction model for stroke volume index decrease based on electrical cardiometry measurements. The content and the scope of the paper are interesting and the paper can be accepted in the present form but for a section in the introduction (lines 89-101) that should be moved into the methods and discussion sections. 

Author Response

We thank you for your comment and we changed the manuscript accordingly.

Reviewer 4 Report

The authors compared routine hemodynamics between transthoracic echo-Doppler and electrical cardiometry by external electrodes in volunteers in whom progressive negative pressures were exerted to the lower limbs, which equalizes blood withdrawal of 10, 10-20 and more than 20 % of total blood volume, respectively.

Strong correlations were found between the different hemodynamic characteristics to a decrease in stroke volume, as induced by the negative pressures.

Comments and remarks

Stroke volume is estimated by TTE by means of VTI x diameter LVOT. The  biggest deviation in measurements is obtained when LVOT diameters are included in the estimation. VTI is a perfect match of stroke volume. Thus, why including another and highly variable parameter?

Changes of CO is much more important than real values. If including VTI instead of stroke volume, one can perfectly describe the degree of change rather than the real values, which could lead to stronger data and results than the change of real values. In fig. 3 diagram G cardiac output is not changing tremendously with increasing negative pressures. Please check if VTI over the aortic valve is used.

Also, the ROC curves could do better if looking to stroke volume (fig. 5). Please check VTI.

L 228: do the authors mean individuals instead of populations?

Author Response

We thank the reviewer for that important comment. The medical literature shows a high number of studies for haemodynamic monitoring and therapy based on SV and / or CO, but as far as we know there is no study optimizing haemodynamics based on VTI or just describing VTI as guiding parameter. Although we fully appreciate and agree with your point, we decided to relate all to the clinically more relevant SV to present the data according to the main body of medical publications.

In this study we measured LVOT diameter at certain points during the protocol and we saw that the LVOT diameter did not change relevantly during the experiment. Our concern was that the LVOT diameter reduces due to a reduced filling state, but that wasn’t the case.

Consequently, we decided to use the initial LVOT diameter measurement for all subsequent individual SV determinations at all other time points and that is exactly as using the VTI as you proposed.

In relation to figure 3 G we would like to comment that the changes of stroke volume were at the first stage added by the decreased heart rate and in the subsequent stages counteracted by the increase of heart rate. This led to the point that the overall changes of CO are not so step-by-step as for SVI.

Regarding the measurement of VTI we used the point in the apical 4-C view corresponding to the measurement site of LVOT diameter in the parasternal short axis view just below the valve area. For every individual we additionally didn’t move the positioning of the volunteer in between measurements of time points and had the same access points for the echocardiographic measurements to further reduce variability of measurements.

As mentioned above we appreciate the reviewers remark and modified the description of the LVOT measurement to the manuscript.

Additionally:

We performed 1000 times bootstrapping and we agree with the reviewer that we should consequently modify the sentence. We changed that to bootstrap samples instead of populations.

Reviewer 5 Report

This is an original interesting study, the manuscript being well written. The theme of this study, including machine-learning algorithms, take part of a large field of research. As cardiologist, I consider that the protocol of echocardiographic measurements is correct. The results are promising. Therefore, I consider that this manuscript deserve to be published.

Author Response

We thank the reviewer for that comment.